# You Shall not Pass: the Zero-Gradient Problem in Predict and Optimize for Convex Optimization

## Abstract

Predict and optimize is an increasingly popular decision-making paradigm that employs machine learning to predict unknown parameters of optimization problems. Instead of minimizing the prediction error of the parameters, it trains predictive models using task performance as a loss function. In the convex optimization domain, predict and optimize has seen significant progress due to recently developed methods for differentiating optimization problem solutions over the problem parameters. This paper identifies a yet unnoticed drawback of this approach – the zero-gradient problem – and introduces a method to solve it. The suggested method is based on the mathematical properties of differential optimization and is verified using two real-world benchmarks.

## 1 Introduction

Mathematical programming is one of the fundamental tools of applied mathematics. It is utilized in various domains, such as finance [Cornuejols and Tütüncü, 2006], power systems [Bansal, 2005], robotics [Raja and Pugazhenthi, 2012], and many others. The main practical limitation of mathematical programming is that it requires a fully-defined model describing the problem which is not always available in reality. A promising approach to overcome this limitation is to employ machine learning (ML) to predict missing parts of the model [Ning and You, 2019].

*Predict and optimize* (P&O) [Elmachtoub and Grigas, 2017] is a decision-making paradigm that combines ML with mathematical programming. It considers optimization problems where some parameters are unknown and should be predicted prior to solving the problem. The P&O approach builds upon the observation that naively training an ML algorithm to match the distribution of unknown parameters is inefficient [Elmachtoub and Grigas, 2017], as this approach does not take the actual task performance into account. Instead, P&O aims at using task performance as the objective function for ML models directly.

The standard approach to training models in machine learning is to use gradient-based algorithms, such as stochastic gradient descent Kiefer and Wolfowitz [1952]. In predict and optimize, computing the gradient of the task performance involves differentiating the solution of the optimization problem with respect to the parameters, which is a non-trivial task. In their seminal work, Agrawal et al. [2019a] have shown that a large class of convex optimization problems indeed can be differentiated.

In this paper, we identify a fundamental drawback of differential optimization – the *zero-gradient problem*. Specifically, we show that the Jacobian of convex problems often has a large null space, and hence the task performance, as a function of the ML model parameters, is flat in a significant part of its domain. Therefore, it can not be optimized using gradient-based methods. Consequently, we introduce a way to compute an approximate gradient that is zero only in the optimal solution and is

guaranteed to not decrease performance. Finally, we validate the superiority of this method using two real-world problems: the portfolio optimization problem and the optimal power flow problem.

## 2 Predict and optimize

In this section, we provide an overview of the existing research in the predict and optimize domain. Then, we define the P&O problem and introduce the solution approach that we are going to investigate later in this manuscript.

### 2.1 Related work

To the best of our knowledge, the predict and optimize framework was first introduced by Elmachtoub and Grigas [2017]. They consider optimization problems with linear objectives and derive a convex approximation of the task performance function. Then, they optimize the prediction model by using sub-gradients of this approximation. Later, this method was extended onto combinatorial problems by Mandi et al. [2020]. Several other approximations were introduced in other studies focusing on combinatorial problems. Vlastelica et al. [2019] derive a differentiable piecewise-linear approximation for the task performance; Berthet et al. [2020] employ stochastic perturbations to approximate derivative of combinatorial problems.

Unlike in the combinatorial case, continuous convex optimization problems do allow exact differentiation of the loss function. The sequence of works [Amos and Kolter, 2017], [Agrawal et al., 2019b], [Agrawal et al., 2019a] developed a differential optimization technique to compute the derivative of convex optimization problems. In their latest work [Agrawal et al., 2019a], the authors delivered a general method that allows differentiating disciplined convex programs [Grant et al., 2006]. This result gave rise to new applications of P&O to convex optimization: Uysal et al. [2021] applied convex differential optimization to the risk budgeting portfolio optimization problem; Wang et al. [2020] utilized it to learn surrogate models for predict and optimize; Donti et al. [2017] applied the method to three different real-world benchmarks. Moreover, several studies applied differential optimization to predict and optimize for other problem classes. In Wilder et al. [2019], it was used in linear optimization via constructing a quadratic approximation of the problem. Later, Mandi and Guns [2020] improved upon this result by using logarithmic approximations. Ferber et al. [2020] combined a similar idea with the cutting plane approach and used differential optimization in combinatorial problems.

Outside of predict and optimize, differential optimization also has found several applications. Chen et al. [2021] used it to train reinforcement learning agents in the action space with convex constraints, and Agrawal et al. [2019c], employed it for tuning model predictive control algorithms.

While the benefits of the differential optimization approach to predict and optimize are numerous, it is still not fully understood. It was reported in several studies Vlastelica et al. [2019], Wilder et al. [2019], that the gradient of a linear problem is zero everywhere, except for the finite set of points where it is undefined. Since any linear problem is convex, this observation suggests that the gradients of convex problems should be also thoroughly investigated.

### 2.2 Problem formulation

In this section, we introduce the P&O problem. We refer readers to Elmachtoub and Grigas [2017] for further details. In predict and optimize, we solve optimization problems of the form

$$\arg\max_x f(x, w) \text{ s. t. } x \in \mathcal{C}, \qquad \text{(True problem)}$$

where $x \in \mathbb{R}^n$ is the decision variable, $w \in \mathbb{R}^u$ is a vector of unknown parameters, $f : \mathbb{R}^n \times \mathbb{R}^u \to \mathbb{R}$ is the objective function, and $\mathcal{C}$ is the feasibility region. The defining feature of this problem is that the parameters $w$ are unknown at the moment when the decision must be made. Therefore, the true optimization problem is under-defined and cannot be solved directly.

One way to deal with the unknown parameters $w$ is to use a prediction $\hat{w}$ instead. Then, the decision can be computed by solving the following problem, which we refer to as the internal problem:

$$x^*(\hat{w}) = \arg\max_x f(x, \hat{w}) \text{ s. t. } x \in \mathcal{C}. \qquad \text{(Internal problem)}$$

A commonly made assumption is that instead of $w$, we observe a feature vector $o$ that contains some information about $w$. Also, we have a dataset $\mathcal{D} = \{(o_k, w_k)\}$, e.g., of historical data, which we can use to learn the relation between $w$ and $o$. This setup enables using ML models to compute the prediction. We denote the prediction model by $\phi_\theta$, and thus we have $\hat{w} = \phi_\theta(o)$.

The problem described above is not specific to predict and optimize. What separates the P&O paradigm from earlier works is the approach to training the model $\phi_\theta$. In the past, machine learning models would be trained to predict $w$ as accurately as possible, e.g., in Mukhopadhyay and Vorobeychik [2017]. However, the parameter prediction error is merely an artificial objective and our true goal is to derive a decision $x$ that maximizes the task performance $f(x, w)$. The main goal of the P&O approach is to utilize this objective for training the model $\phi_\theta$. The task performance achieved by $\phi_\theta$ on the dataset $\mathcal{D}$ can be quantified by the following loss function:

$$L(\theta) = -\frac{1}{|\mathcal{D}|} \sum_{(o,w) \in \mathcal{D}} f\left(x^*\big(\phi_\theta(o)\big), w\right) \tag{1}$$

Most machine learning algorithms for training models are based on computing the gradient of the loss function (Kiefer and Wolfowitz [1952]). To train $\phi_\theta$ with a gradient-based algorithm, we need to differentiate $L$ over $\theta$, and hence we need to compute the gradient $\nabla_\theta f\left(x^*(\hat{w}), w\right)$, where $\hat{w} = \phi_\theta(o)$. Applying the chain rule, it can be decomposed into three terms:

$$\nabla_\theta f\left(x^*(\hat{w}), w\right) = \nabla_x f\left(x^*(\hat{w}), w\right) \nabla_{\hat{w}} x^*(\hat{w}) \nabla_\theta \hat{w}. \tag{2}$$

The second term, $\nabla_{\hat{w}} x^*(\hat{w})$, is the Jacobian of the solution of the optimization problem over the prediction $\hat{w}$. An exact method to compute this Jacobian was introduced in Agrawal et al. [2019a], but it has never been thoroughly analyzed. In the next section, we show that $\nabla_{\hat{w}} x^*(\hat{w})$ has a large null space, thereby causing the total gradient in Eq. 2 to be zero even outside of the optimum.

## 3 Differentiable optimization

In this section, we study the derivative of convex optimization programs over the parameters of the objective function. We show that the gradient in Eq. 2 is often zero outside of the optimum, and hence it causes gradient-following methods to get stuck in suboptimal solutions. In the second part of this section, we introduce a method to solve this problem.

Without loss of generality, we consider a single instance of the problem, i.e., one sample $(o, w) \in \mathcal{D}$. Everywhere in this section, we denote the prediction by $\hat{w} = \phi_\theta(o)$. Then, the decision is computed as a solution of the internal optimization problem defined as follows:

$$x^*(\hat{w}) = \arg\max_x f(x, \hat{w}) \text{ s.t. } x \in \mathcal{C}. \tag{3}$$

We use $\hat{x}$ to denote the value of $x^*(\hat{w})$ for a given prediction $\hat{w}$. As we are interested in convex optimization problems, we make the following assumptions:

**Assumption 1.** *The objective function $f(x, w)$ is concave and twice continuously differentiable in $x$ for any $w$.*

**Assumption 2.** *The feasibility region $\mathcal{C}$ is convex, i.e., $\{\mathcal{C} = \{x | g_i(x) \leq 0, i = 1, \ldots, l\}$, where $g_i(x)$ are convex differentiable functions. Moreover, for any $x \in \mathcal{C}$, the gradients $\{\nabla_x g_i(x) | g_i(x) = 0\}$ of the active constraints are linearly independent.* [1]

Additionally, we make an assumption about how $f$ depends on $w$, which holds for many real-world problems, including linear and quadratic optimization problems.

**Assumption 3.** *The objective function $f(x, w)$ is twice continuously differentiable in $w$.*

Throughout this paper, we use derivatives of different objects. For clarity, we first provide an overview of them: the gradient of the true objective function over the decision, $\nabla_x f(\hat{x}, w)$; the Jacobian of the decision over the prediction, $\nabla_{\hat{w}} x^*(\hat{w})$; the Jacobian of the prediction over the ML model parameters, $\nabla_\theta \hat{w}$; and the gradient of the predicted objective in the internal problem, $\nabla_x f(x, \hat{w})$. In the next section, we establish some crucial properties of the Jacobian $\nabla_{\hat{w}} x^*(\hat{w})$.

---

[1] As is, Assumption 2 does not allow equality constraints. For clarity, we use this formulation in the main body of the paper. In the appendix, we show that our results hold for the equality constraints as well.

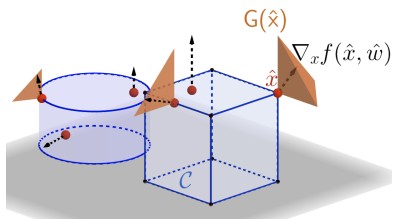

Figure 1: Gradient cones $\hat{x} + G(\hat{x})$ (orange cones) and internal gradients $\nabla_x f(\hat{x}, \hat{w})$ (black arrows) at different points $\hat{x}$ (red dots) in different feasibility regions $\mathcal{C}$ (blue cube and cylinder).

### 3.1 The zero-gradient theorem

We begin by investigating the relation between the values of the function $x^*(\hat{w})$ and the gradient of the internal objective, $\nabla_x f(x, \hat{w})$. Let $n_i := \nabla_x g_i(\hat{x})$, $i = 1, \ldots, l$ be the normal vectors of the constraints at $\hat{x}$, Then, the KKT conditions Kuhn and Tucker [1951] at $\hat{x}$ state that there exist real values $\alpha_1, \ldots, \alpha_l$ such that the following holds:

$$\nabla_x f(\hat{x}, \hat{w}) = \sum_{i=1}^{l} \alpha_i n_i, \quad \alpha_i g_i(\hat{x}) = 0, \quad \alpha_i \geq 0, \quad g_i(\hat{x}) \leq 0, \quad i = 1, \ldots, l.$$

Under Assumptions 1 and 2, the KKT multipliers $\alpha_i$ are uniquely defined by $\hat{w}$ and $\hat{x}$. Thus, as $\hat{x}$ is defined by $\hat{w}$, we sometimes write $\alpha_i(\hat{w})$ to emphasize that it is, in fact, a function of $\hat{w}$. To provide a geometrical perspective on the KKT conditions, we introduce the following definition:

**Definition 3.1.** *Let $x \in \mathcal{C}$ and let $I(x) = \{i | g_i(x) = 0\}$ be the set of indices of the constraints active at $x$. Let $n_i = \nabla_x g_i(x)$, $\forall i \in I(x)$, be the normal vectors of these constraints. The gradient cone, $G(x) := \left\{ \sum_{i \in I} \alpha_i n_i | \alpha_i \geq 0 \right\}$, is the positive linear span of normal vectors $n_i$.*

Combining the KKT conditions with Definition 3.1, we immediately arrive at the following property:

**Property 3.2.** *Let $x \in \mathcal{C}$ and let $\nabla_x f(x, \hat{w})$ be the internal gradient at $x$. Then, $x$ is a solution to the problem in Eq. 3 if and only if $\forall i \in I(x), \exists \alpha_i \geq 0$, such that $\nabla_x f(x, \hat{w}) = \sum_{i \in I(x)} \alpha_i n_i \in G(x)$, where $I(x)$ is the set of indices of active constraints, $I(x) = \{i | g_i(x) = 0\}$.*

While trivial, this property provides a geometrical interpretation of the problem. Effectively, a point $x$ is a solution to the problem in Eq. 3 if and only if the internal gradient at this point lies inside its gradient cone. Figure 1 illustrates this property.

Before studying the Jacobian $\nabla_{\hat{w}} x^*(\hat{w})$, we first need to address the question of when this Jacobian exists. Sufficient conditions for existence are given in Fiacco [1976]. Under Assumptions 1-3, these conditions can be reformulated as follows:

**Lemma 3.3** (Theorem 2.1 in Fiacco [1976]). *Let Assumptions 1-3 hold and let*

$$\nabla_x f(\hat{x}, \hat{w}) = \sum_{i \in I(\hat{x})} \alpha_i(\hat{w}) n_i$$

*be the representation of the internal gradient with the normals of the active constraints. Then, suppose that the strict complementary slackness condition holds, i.e., $\alpha_i(\hat{w}) > 0$, $\forall i \in I(\hat{x})$. Then, the Jacobian $\nabla_{\hat{w}} x^*(\hat{w})$ exists at $\hat{w}$. Moreover, $\alpha_i(\cdot)$ is continuous around $\hat{w}$ for any $i \in I(\hat{x})$.*

Proof of this lemma is given in Fiacco [1976]. This result establishes that strict complementary slackness is sufficient for the Jacobian $\nabla_{\hat{w}} x^*(\hat{w})$ to exist. In most cases, the points that violate strict complementary slackness form a zero-measure set and hence can be neglected in practice.

Now, we have all the necessary tools to describe the structure of the Jacobian $\nabla_{\hat{w}} x^*(\hat{w})$. Suppose that the strict complementary slackness condition holds at $\hat{x}$ and hence the Jacobian exists. Assume that we perturb $\hat{w}$ and obtain $\hat{w}'$. Let $\hat{x}' = x^*(\hat{w}')$ denote the solution corresponding to $\hat{w}'$. What can be said about $\hat{x}'$? Strict complementary slackness implies that the constraints active at $\hat{x}$ will remain active at $\hat{x}'$ if the difference $\|\hat{w}' - \hat{w}\|_2^2$ is small enough. Therefore, the decision $\hat{x}'$ can only move within the tangent space of $\mathcal{C}$ at $\hat{x}$, i.e., orthogonally to all $n_i$, $i \in I(\hat{x}.)$ Hence, when more constraints are active, $\hat{x}'$ can move in less directions. Formally, we obtain the following lemma:

**Lemma 3.4.** *Suppose that the strict complementary slackness conditions hold at $\hat{x}$ and let $\nabla_x f(\hat{x}, \hat{w}) = \sum_{i \in I(\hat{x})} \alpha_i n_i$, $\alpha_i > 0$, $\forall i \in I(\hat{x})$ be the internal gradient. Let $\mathcal{N}(\hat{x}) = span(\{n_i \,|\, i \in I(\hat{x})\})$ be the linear span of the gradient cone. Then $\mathcal{N}(\hat{x})$ is contained in the left null space of $\nabla_{\hat{w}} x^*(\hat{w})$, i.e., $v \, \nabla_{\hat{w}} x^*(\hat{w}) = 0$, $\forall v \in \mathcal{N}(\hat{x})$*

The formal proof of this result can be found in the appendix. Lemma 3.4 is very important, as it specifies in what directions $x^*(\hat{w})$ *can move* as a consequence of changing $\hat{w}$. Now, the first term in the chain rule in Eq. 2, $\nabla_x f(\hat{x}, w)$, specifies in what directions $x^*(\hat{w})$ *should* move in order for the true objective to increase. Naturally, if these directions are contained in the null space of $\nabla_{\hat{w}} x^*(\hat{w})$, then the total gradient in Eq. 2 is zero. This observation constitutes the main theorem of this paper – the zero-gradient theorem.

**Theorem 3.5** (Zero-gradient theorem)**.** *Let $\hat{w}$ be a prediction, and let $\hat{x}$ be the solution of the internal optimization problem defined in Eq. 3. Suppose that the strict complementary slackness conditions hold at $\hat{x}$ and let $\mathcal{N}(\hat{x}) = span(\{n_i \,|\, i \in I(\hat{x})\})$ be the linear span of the gradient cone at $\hat{x}$. Then, $\nabla_x f(\hat{x}, w) \in \mathcal{N}(\hat{x}) \implies \nabla_\theta f(\hat{x}, w) = 0$.*

The proof of this theorem is obtained by simply applying Lemma 3.4 to the chain rule in Eq. 2. The theorem claims that the gradient of the P&O loss in Eq. 1 can be zero in the points outside of the optimal solution. Hence, any gradient-following method "shall not pass" these points. In particular, the zero-gradient phenomenon happens in such points $\hat{x}$ where the true gradient $\nabla_x f(\hat{x}, w)$ is contained in the space $\mathcal{N}(\hat{x})$ spanned by the gradient cone $G(\hat{x})$. As the dimensionality of this space grows with the number of active constraints, the zero-gradient issue is particularly important for problems with a large number of constraints. In the worst case, $\mathcal{N}(\hat{x})$ can be as big as the whole decision space $\mathbb{R}^n$, thereby making the total gradient $\nabla_\theta f(\hat{x}, w)$ from Eq. 2 zero for any value of the true gradient $\nabla_x f(\hat{x}, w)$. In the following sections, we introduce a method that resolves the zero-gradient problem and provides theoretical guarantees for its performance.

## 3.2 Quadratic programming approximation

The fundamental assumption of the predict and optimize framework is that training $\phi_\theta$ using the task performance loss is better than fitting it to the true values of $w$. Hence, the models trained with predict and optimize might output $\hat{w}$ that is significantly different from the true $w$ and yet produces good decisions. Taking this argument one step further, we claim that the objective function $f(x, \hat{w})$ in the internal optimization problem in Eq. 3 does not need to be the same as the true objective $f(x, w)$. In particular, we suggest computing decisions using a simple quadratic program (QP):

$$x^*_{QP}(\hat{w}) = \arg\max_x -\|x - \hat{w}\|_2^2 \text{ s.t. } x \in \mathcal{C}. \tag{4}$$

The reasons for this choice are manifold. First, the internal objective $f_{QP}(x, \hat{w}) = -\|x - \hat{w}\|_2^2$, is strictly concave and hence $x^*_{QP}(\hat{w})$ is always uniquely-defined. Moreover, the range of $x_{QP}(\hat{w})$ is $\mathcal{C}$, i.e., $\forall x \in \mathcal{C}$, $\exists \hat{w}$ such that $x = x^*_{QP}(\hat{w})$. Hence, it can represent any optimal solution. However, the most important property of QP is that its Jacobian is very simple, which we explain below.

The problem in Eq. 4 has a simple geometrical interpretation: the point $x = \hat{w}$ is the unconstrained maximum of $f_{QP}(x, \hat{w})$ and $x^*_{QP}(\hat{w})$ is its Euclidean projection on the feasibility set $\mathcal{C}$, see Figure 2. To compute the Jacobian $\nabla_{\hat{w}} x^*_{QP}$, we need to understand how perturbations of $\hat{w}$ affect $x^*_{QP}$. Employing the geometrical intuition above, we obtain the following lemma:

**Lemma 3.6.** *Let $\hat{w}$ be a prediction and $\hat{x}$ be the optimal solution of the QP problem defined in Eq. 4. Let the strict complementary slackness condition hold and let $\{n_i | i \in I(\hat{x})\}$ be the normals of the active constraints. Let $\{e_j | j = 1, \ldots, n - |I(\hat{x})|\}$ be an orthogonal complement of vectors $\{n_i | i \in I(\hat{x})\}$ to a basis of $\mathbb{R}^n$. Then, the representation of the Jacobian $\nabla_{\hat{w}} x_{QP}(\hat{w})$ in the basis $\{n_i\} \cup \{e_j\}$ is a diagonal matrix. Its first $|I(\hat{x})|$ diagonal entries are zero, and the others are one.*

Proof of this lemma can be found in the appendix. Lemma 3.6 implies that the Jacobian $\nabla_{\hat{w}} x_{QP}(\hat{w})$ has a simple form and can be easily computed by hand. While providing computational benefits, this approach does not address the zero-gradient problem. In the next section, we introduce a method to compute an approximate of the Jacobian $\nabla_{\hat{w}} x_{QP}(\hat{w})$ that has a strictly one-dimensional null space. Combined with the QP approximation, it is guaranteed to at least not decrease the task performance.

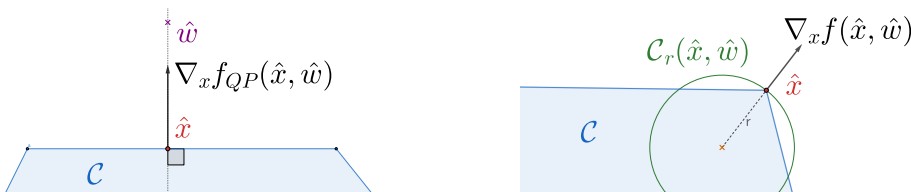

Figure 2: *Left*: Illustration of QP. The internal gradient (black arrow) at the solution of the QP $\hat{x}$ (red point) is orthogonal to the feasibility region $\mathcal{C}$ (blue area) and points towards the unconstrained maximum $\hat{w}$ (purple cross). *Right*: Illustration of the $r-$smoothed problem. The internal gradient (black arrow) is orthogonal to the $r-$smoothed feasibility region $\mathcal{C}_r(\hat{x}, \hat{w})$ (green circle) at the decision $\hat{x}$ (red point).

### 3.3 Local smoothing

We identified a fundamental issue of differential optimization – the zero-gradient problem. We showed that the null space of the Jacobian $\nabla_{\hat{w}} x(\hat{w})$ depends on the number of constraints active at $\hat{x}$. Generally, this number can be as large as the number of optimized variables $n$, and the gradient-descent algorithms can get stuck in certain points on the boundary of the feasibility region.

In this section, we propose a simple way to modify the feasibility region – we smooth $\mathcal{C}$ locally around the point for which we compute the Jacobian, thereby ensuring that its null space becomes one dimensional. First, we define a method for the general setup, without imposing any assumptions on the optimization problem. Then, we demonstrate that combined with the QP approximation from Section 3.2, this smoothing approach has theoretical guarantees.

We begin with the general case – the problem in Eq. 3. Let $\nabla_x f(\hat{x}, \hat{w}) = \sum_{i \in I(\hat{x})} \alpha_i n_i$ be the internal gradient at $\hat{x}$ for some $\alpha_i \geq 0$, $\forall i \in I(\hat{x})$. Then, we introduce the following definition:

**Definition 3.7.** *Let $r > 0$ be a positive real number. Let $c = \hat{x} - r \frac{\nabla_x f(\hat{x}, \hat{w})}{\|\nabla_x f(\hat{x}, \hat{w})\|_2}$. The local $r$-smoothed feasibility region, $\mathcal{C}_r(\hat{x}, \hat{w}) := \{y | y \in \mathbb{R}^n, \|y - c\|_2 \leq r\}$, is a ball of radius $r$ around $c$. The local $r-$smoothed problem $P_r(\hat{x}, \hat{w})$ with parameters $\hat{x}, \hat{w}$ is defined as $x_r^*(\hat{w}) := \arg\max_{x \in \mathcal{C}_r(\hat{x}, \hat{w})} f(x, \hat{w})$.*

Figure 2 shows an example of the local $r-$smoothed problem. Now, let $\hat{x}_r = x_r^*(\hat{w})$ denote the solution of $P_r(\hat{x}, \hat{w})$. By construction, the internal gradient at $\hat{x}_r$ lies in the one-dimensional gradient cone, and hence, by Property 3.2, $\hat{x}_r = \hat{x}$. The main purpose of smoothing is to approximate the gradient in Eq. 2 by substituting $\nabla_{\hat{w}} x^*(\hat{w})$ with $\nabla_{\hat{w}} x_r^*(\hat{w})$. We highlight that the decisions are still computed using the non-smoothed problem $x^*(\hat{w})$ and $x_r^*(\hat{x}, \hat{w})$ is used exclusively to perform the gradient update step. In other words, we use the following expression to compute the gradient:

$$\nabla_\theta f(x^*(\hat{w}), w) \approx \nabla_x f(\hat{x}, w) \, \nabla_{\hat{w}} x_r^*(\hat{w}) \, \nabla_\theta \hat{w} \tag{5}$$

It is worth mentioning that the strict complementary slackness in the original problem is a stronger condition than the strict complementary slackness on $P_r(\hat{x}, \hat{w})$. Therefore, the Jacobian of the $r-$smoothed problem can exist even for predictions $\hat{w}$ where the true Jacobian does not.

Generally, the efficiency of $r-$smoothing depends on the form of the internal problem in Eq. 3. Below, we show that combining $r-$smoothing with the QP approximation has guarantees on its performance. First, we notice that Lemma 3.6 prescribes the Jacobian of the $r-$smoothed QP problem:

**Property 3.8.** *Let $\hat{x} = x_{QP}^*(\hat{w})$ be a decision derived via QP. Suppose that the complementary slackness conditions hold for $P_r(\hat{x}, \hat{w})$ and let $e_1 = \nabla_x f_{QP}(\hat{x}, \hat{w})$ be the internal gradient. Let $\{e_2, \ldots, e_n\}$ be a complement of $e_1$ to an orthogonal basis of $\mathbb{R}^n$. Then, the Jacobian $\nabla_{\hat{w}} x_r^*(\hat{w})$ of the local $r-$smoothed problem expressed in the basis $\{e_1, e_2, \ldots, e_n\}$ is a diagonal matrix. Its first entry is zero, others are ones.*

As $\mathcal{C}_r(\hat{x}, \hat{w})$ is defined by a single constraint, the null space of $\nabla_{\hat{w}} x_r^*(\hat{x}, \hat{w})$ is always one-dimensional. Hence, the zero-gradient problem can only occur when the internal gradient $\nabla_x f_{QP}(\hat{x}, \hat{w})$ and the true gradient $\nabla_x f(\hat{x}, w)$ are exactly collinear. Hence, we expect $r-$smoothing to significantly improve upon the zero-gradient problem. Next, we show that the $r-$smoothed Jacobian is actually a good approximation. In the following theorem, we demonstrate that the local $r-$smoothing of the QP approach indeed yields a "good" direction for the gradient steps.

**Theorem 3.9.** *Let $\hat{x} = x_{QP}^*(\hat{w})$ be the decision obtained via QP and let $\nabla_{\hat{w}} x_r^*(\hat{w})$ be the Jacobian of the $r-$smoothed QP problem. Let $\Delta \hat{w} = \nabla_x f(\hat{x}, w) \nabla_{\hat{w}} x_r^*(\hat{w})$ be the prediction perturbation obtained by using this Jacobian and let $\hat{w}'(t) = \hat{w} + t \Delta \hat{w}$ be the updated prediction. Then, for $t \to 0^+$, using $\hat{w}'(t)$ results in a non-decrease in the task performance. In other words, $f\big(x_{QP}^*(\hat{w}'(t)), w\big) \geq f\big(x_{QP}^*(\hat{w}), w\big)$.*

Interestingly, this result does not depend on $r$. However, this is to be expected – no matter the radius of $\mathcal{C}_r$, the Jacobian of $P_r(\hat{x}, \hat{w})$ is still the same by Lemma 3.6. Theorem 3.10 shows that using $r-$smoothing together with the QP approximation results in analytically computable Jacobian that has a strictly one-dimensional null space. Therefore, we are much less likely to encounter the zero-gradient problem when using this approximation. However, the resulting one-dimensional null space contains the only direction that can move the prediction $\hat{w}$, and hence the decision $\hat{x}$, inside $\mathcal{C}$. This might become crucial, for example, when the optimal solution with respect to the true objective lies in the interior of $\mathcal{C}$. To resolve this problem, we use the projection distance regularization method first suggested in Chen et al. [2021]. Specifically, we add a penalty term

$$p(\hat{w}) = \alpha \|\hat{x} - \hat{w}\|_2^2, \tag{6}$$

where $\alpha \in \mathbb{R}^+$ is a hyperparameter. Minimizing this term, we push $\hat{w}$ along the null-space of the Jacobian towards the feasibility region and eventually move $\hat{x}$ inside $\mathcal{C}$.

### 3.4 The training process

In this section, we summarize the results of Sections 3.1-3.3 and describe the final algorithm we use to solve the P&O problems. For each problem instance $(o, w)$, we first compute the prediction, $\hat{w} = \phi_\theta(o)$, and the decision using the QP approximation method, $\hat{x} = x_{QP}^*(\hat{w})$. Then, we obtain the achieved objective value, $f(\hat{x}, w)$. During training, we update the model parameters $\theta$ by performing the steps described in Algorithm 1.

---

**Algorithm 1**

---

**for** $(o, w) \in \mathcal{D}$ **do**

$\quad \hat{x} \leftarrow x_{QP}^*\big(\phi(o)\big)$ $\qquad\qquad\qquad\qquad\qquad\qquad\qquad\qquad\qquad$ ▷ Compute the decision

$\quad f_x \leftarrow \nabla_x f(\hat{x}, w)$ $\qquad\qquad\qquad\qquad\qquad\qquad\qquad\qquad\quad$ ▷ Compute the true gradient

$\quad \hat{f}_x \leftarrow \nabla_x f(\hat{x}, \hat{w})$ $\qquad\qquad\qquad\qquad\qquad\qquad\qquad\qquad$ ▷ Compute the internal gradient

$\quad f^0 \leftarrow \frac{f_x^\top \hat{f}_x}{\|\hat{f}_x\|_2}$ $\qquad\qquad\qquad$ ▷ Project the true gradient on the null space of $\nabla_{\hat{w}} x_r^*(\hat{w})$

$\quad \Delta \hat{w} \leftarrow f_x \nabla_{\hat{w}} x_r^*(\hat{w}) = f_x - f^0.$ $\qquad\qquad\qquad\quad$ ▷ Compute the prediction perturbation

$\quad \Delta \hat{w}^{reg} \leftarrow 2\alpha(\hat{x} - \hat{w})$ $\qquad\qquad\qquad\quad$ ▷ Compute the anti-gradient of the penalty from Eq. 6

$\quad \Delta \theta \leftarrow (\Delta \hat{w} + \Delta \hat{w}^{reg}) \nabla_\theta \phi_\theta(o)$ $\qquad\qquad\qquad\qquad$ ▷ Approximate the total gradient

$\quad \theta \leftarrow \theta + \eta \Delta \theta$ $\qquad\qquad\qquad\qquad\qquad\qquad$ ▷ Perform the gradient step of size $\eta$

---

## 4 Experiments

The main result of Section 3 is the zero-gradient theorem, which describes when the gradient $\nabla_\theta f\big(x^*(\hat{w}), w\big)$ is zero. To deal with it, we introduced the QP approach for computing the decisions, $r-$smoothing for approximating the Jacobian $\nabla_{\hat{w}} x^*(\hat{w})$, and projection distance regularization to deal with the remaining null space dimension. Our solution deals with the zero gradient problem by combining these methods. In this section, we use two real-world P&O problems to evaluate the efficiency of our method.

### 4.1 Portfolio optimization

Following Wang et al. [2020], we apply the predict and optimize framework to the Markowitz mean-variance stock market optimization problem Markowitz and Todd [2000]. In this problem, we act as an investor who seeks to maximize the immediate return but minimize the risk penalty. The

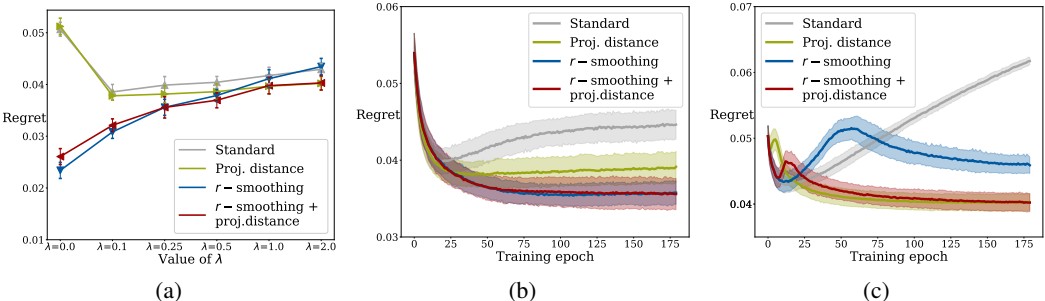

(a)            (b)            (c)

Figure 3: Comparison of different methods on the portfolio optimization problem. $y-$axis represents the mean and the standard deviation of the regret on the test set for four seeds. The lower the better. *(a)* Regret for different $\lambda$. *(b)* Regret during training for $\lambda = 0.25$. *(c)* Regret during training for $\lambda = 2$.

decision variable, $x \in \mathbb{R}^n$, is a positive vector representing our investment in different securities. The budget constraint forces the investments to add up to one, i.e., $\sum_i x_i = 1$. The objective is defined as $f(x, p, Q) = p^\top x - \lambda x^\top Q x$, where $p \in \mathbb{R}^n$ is the immediate return of the securities, $\lambda \geq 0$ is the risk-aversion weight, and $Q \in \mathbb{R}^{n \times n}$ is the positive definite matrix of covariance between securities. The portfolio optimization problem is then defined as follows:

$$\arg\max_x \ \underbrace{p^\top x - \lambda \, x^\top Q x}_{f(x,p,Q)} \qquad \text{s. t.} \qquad \sum_{i=1}^n x_i = 1, \quad x \geq 0. \tag{7}$$

This is a quadratic optimization problem with unknown parameters $(p, Q)$, as neither the immediate return nor the true covariance matrix is known at the decision-making moment. Following Wang et al. [2020], we use historical data from QUANDL WIKI prices QUANDL [2020] for 505 largest companies on the American market for the period 2014-2017. The dataset is processed and for every day we obtain a feature vector summarizing the recent price dynamic. For further details on the processing we refer readers to the code[2] and to Wang et al. [2020]. For each run, we randomly split data into train, validation, and test sets by using 70%, 15%, and 15% of the whole dataset respectively. To evaluate the performance of different algorithms, we use *regret,* defined as

$$\text{regret}(o, w) = f\Big(x^*\big(\phi_\theta(o), w\big)\Big) - \max_x f\big(x, w\big). \tag{8}$$

In the experiments, we used $\lambda$ from the set $\{0, 0.1, 0.25, 0.5, 1, 2\}$. For the larger value of $\lambda$, the true objective $f(x, p, Q)$ is "more" quadratic, and hence its maximum is more likely to lie in the interior of $\mathcal{C}$. For smaller $\lambda$'s, on the other hand, the true objective becomes almost linear and hence it usually attains its maximum on the boundary of $\mathcal{C}$.

First of all, we define the QP approximation of the portfolio optimization problem:

$$x^*(\hat{w}) = \arg\max_x -(x - \hat{w})^\top I (x - \hat{w}) \qquad \text{s. t.} \qquad \sum_{i=1}^n x_i = 1, \quad x \geq 0. \tag{9}$$

As the problem is quadratic, the only difference introduced by the QP approximation comes from using the identity matrix $I$ instead of $Q$. The results in Table 1 indicate that it performs at least as well as learning to predict both $p$ and $Q$, and hence in all other experiments we used the QP approximation to compute the decisions.

To investigate the zero-gradient effect, we compared four ways to train the predictor: with/without $r-$smoothing and with/without the penalty term from Eq. 6. The model used for

Table 1: Performance of the QP approximation

| | Final test regret | |
|---|---|---|
| $\lambda$ | QP approximation | Predict both $Q$ and $p$ |
| 0 | **0.061 ± 0.002** | 0.064 ± 0.002 |
| 0.1 | **0.047 ± 0.002** | 0.052 ± 0.002 |
| 0.25 | **0.040 ± 0.002** | 0.045 ± 0.002 |
| 0.5 | **0.039 ± 0.001** | 0.041 ± 0.001 |
| 1 | **0.040 ± 0.002** | 0.041 ± 0.002 |
| 2 | **0.039 ± 0.001** | 0.04 ± 0.001 |

[2]Placeholder for the link to the GitHub repository

the predictor is a 2-layer neural network, further details on the training process are described in the appendix. The results in Figure 3 indicate that $r-$smoothing significantly improves the performance when the true objective is more linear. This result matches the theory from Section 3, as linear true objective pushes the decision $\hat{x}$ towards the boundary of $\mathcal{C}$, and hence it is more likely to enter points with a large gradient cone. For the more quadratic objectives, the true maximum is often in the interior of $\mathcal{C}$, and hence $r-$smoothing alone is not sufficient to reach it. In this case, the regularization term from Eq 6 becomes crucial, as it is the only method that can push $\hat{x}$ inside $\mathcal{C}$.

## 4.2  Optimal power flow in a DC grid

To further understand the zero-gradient phenomenon, we consider the optimal power flow problem (OPF) for DC grids [Li et al., 2018]. Due to power losses, the constraints in this problem are non-linear, thus making it computationally hard. In our experiments, we used a linearized version of the problem that represents a DC grid without power losses. The decision variable is the vector of nodal voltages $v \in \mathbb{R}^n$, and the unknown parameter $w$ represents either the value gained by serving power to a customer or the price paid for utilizing a generator. The reference voltage $v_0 \in \mathbb{R}$, the admittance matrix $Y$, and the constraint bounds represent the physical properties of the grid.

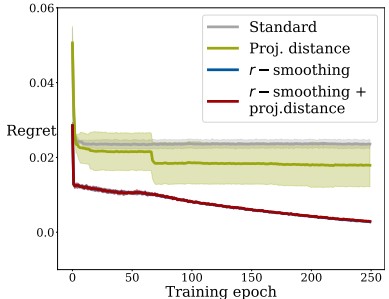

Figure 4: Comparison of different methods on the DC grid OPF problem. $y-$axis represents the mean and standard deviation of the test regret for twelve random seeds.

$$\max_{v} \quad f(v, w) = -v_0 w^\top (Yv)$$
$$\text{subject to:} \quad \underline{V} \leq v \leq \bar{V}$$
$$\underline{P} \leq -v_0 Y v \leq \bar{P}$$
$$\underline{I} \leq Y_{ij}(v_i - v_j) \leq \bar{I}$$

We refer the reader to Li et al. [2018] for further details of the problem. Importantly, the feasibility region is defined by multiple linear constraints, and therefore we expect it to have numerous vertices with large gradient cones. The objective function $f(v, w)$ quantifies the social welfare [Veviurko et al., 2022] generated by all the users of the power grid. Importantly, $f(v, w)$ is linear, and hence its maximum lies on the boundary of the feasibility region.

We compared the same four methods as before on this problem using randomly generated grids with four generators and twelve loads. Same as before, we use QP approximation, $x_{QP}^*(\hat{w}) = \arg\max_{x \in \mathcal{C}}(-\|\hat{w} - x\|_2^2)$, to compute the decisions. The results in Figure 4 confirm our hypothesis – even though we differentiate through a quadratic problem, the linearity of the true objective causes the zero-gradient effect as the decision is pushed towards the boundary of the feasibility region. Then, due to a large number of constraints, it is likely to enter a vertex with a large gradient cone and get stuck there. In this case, $r-$smoothing greatly outperforms the standard differential optimization method from Agrawal et al. [2019a], while the projection distance regularization does not help a lot.

## 5  Conclusion

In this work, we discover and explain the zero-gradient problem in P&O for convex optimization. In particular, we show that the null space of the Jacobian of a convex optimization problem can get arbitrarily large, especially in the case with numerous constraints. This phenomenon prevents gradient-following algorithms from learning optimal solutions in convex P&O problems.

To resolve this issue, we introduce a method to compute an approximation of the Jacobian. It is done by smoothing the feasibility region around the current solution and thereby reducing the dimensionality of the null space to one. We prove that the combination of smoothing with the QP approximation results in the gradient update steps that at least do not decrease the task performance, but often allow to escape the zero-gradient cones. To enable movement along the remaining one-dimensional null space, we add a projection distance regularization term. The suggested method leads to significantly better results for the convex P&O problems that suffer from the zero-gradient problem the most – those with many constraints and with the true optimum lying on the boundary of the feasibility set.

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
