# OpenReview forum: "You Shall not Pass: the Zero-Gradient Problem in Predict and Optimize for Convex Optimization"
_NeurIPS.cc/2023/Conference — Submitted to NeurIPS 2023_

### Official Review · Reviewer_1CVQ · 2023-07-04

**Soundness:** 2 fair
**Presentation:** 1 poor
**Contribution:** 2 fair
**Rating:** 3
**Confidence:** 4

**Summary:**

The paper first characterizes the 'zero-gradient' issue---a challenge associated with learning a model in the 'predict-then-optimize' paradigm---in terms of the number of active KKT constraints of the optimization problem. It then proposes a surrogate optimization problem for which the zero-gradient does not arise and evaluates these surrogates on 2 domains.

**Strengths:**

* The paper addresses an important problem, i.e., zero-gradients in predict-then-optimize.
* The paper proposes a novel surrogate.

**Weaknesses:**

1. **The 'zero-gradient theorem' is not novel:** The paper claims to 'discover and explain the zero-gradient problem in P&O for convex optimization'. However, a number of predict-then-optimize papers acknowledge the zero-gradient issue and propose their own surrogates, e.g., Elmachtoub and Grigas [2017] and Wilder et al. [2019]. In fact, for well-defined LPs (with unique solutions), it is known that the task performance/SPO loss is piecewise constant. In contrast, it is known that if the function and constraints are strongly convex, e.g., the portfolio optimization problem in the experiments with $\lambda > 0$, there are no zero-gradients. As a result, it's not clear why this is the domain that the paper chooses to run experiments on...
1. **The experiments have no baselines:** While the paper does provide a surrogate that does not run into a zero-gradient issue, the bar for publication is typically higher, i.e., that this specific surrogate outperforms others from the literature. As the related work section notes, there are other ways to get around the zero-gradient issue, like creating surrogate problems by adding quadratic/exponential regularization terms, however there are no comparisons to any methods not presented in the paper, not even simple baselines like 2-stage, random and optimal.
1. **The paper is poorly written:** There is almost no information in the about the contributions in the abstract and introduction and, as noted above, the paper does not adequately engage with past work.

### Update (20 Aug 2023)

After the discussion with the authors, reading the other reviews and thinking about this paper more, I find myself still recommending rejection. Here are the reasons:
1. The *submitted* version of the paper lacks any discussion of or comparison to related work, and also overclaims (e.g., "the first to discover the zero-gradient theorem"). While the authors have refined their position significantly in the rebuttals (and much improved their contribution as a result), I believe that (a) there still remain important unanswered questions, and (b) these changes lead to a paper with significantly different claims. I discuss both of these in terms of specific contributions below.
1. **Zero-gradient Theorem:** In the responses, the authors acknowledge that the zero-gradient issue has been known for optimization problems with linear objectives. The modified claim, as I understand it, is that the characterization of zero-gradients in this paper is significantly different/improved from the existing understanding. However, (a) the proof is not particularly novel (imo) because it formalizes existing knowledge in the language of KKT conditions, and moreover (b) it is not clear whether this characterization is significantly more powerful than the existing understanding (specifically, do gradients lie in the null space of the normals of the active constraints when the Jacobian matrix isn't zero, and what does this mean intuitively?). I believe that the strength of this contribution is dependent on the answers to these two (imo) unanswered questions.
1. **r-Smoothing Surrogate:** While avoiding zero-gradient issues is definitely a desirable property for a surrogate, it is by no means a *sufficient* for good performance - a number of papers in the literature do not run into zero-gradient issues. As a result, r-smoothing needs to be better motivated and compared to past work. While the authors have done an admirable job of running experiments in the author response period, there are some papers that do very similar things that the authors have not compared to, e.g., Sahoo et. al. (2022). I think there is work that remains to be done in situating their surrogate in the context of the large body of recent work on PtO.

**Questions:**

1. How does your proposed approach do in comparison to baselines from the literature?

**Limitations:**

There are no limitations discussed; in contrast, I believe this paper over-claims its contributions.

---

> ### Author Rebuttal · Authors · 2023-08-08
>
> **1. The zero-gradient theorem is not novel**
>
> It is indeed well known that differentiating through linear programs is impracticable due to zero/undefined gradients. However, to the best of our knowledge, the zero-gradient problem for *nonlinear* convex optimization was not known before.
> The main theoretical results of our paper, Lemma 3.4 and Theorem 3.5, demonstrate that differentiation of generic convex problems can yield non-informative, zero gradients. In fact, we show that the cause of the zero-gradient problem is not the objective function, but the non-smooth points of the constraints set $\mathcal{C}.$
>
> To additionally reiterate the essence of the zero-gradient problem, we would like to refer to Figure 1. The right half of the figure depicts the constraints set $\mathcal{C}$ defined as a three-dimensional cube. Consider the red point denoted by $\hat{x}$ and the gradient cone $G(x)$ depicted by the orange cone at this point. By Property 3.2 (KKT conditions), the point $\hat{x}$ is the solution of the internal optimization problem, $\arg\max_{x\in\mathcal{C}} f(x,\hat{w})$ iff the gradient of its objective, $\nabla_{x}f(\hat{x}, \hat{w})$, lies in $G(x)$.
> Now, if this gradient lies in the interior of the gradient cone (i.e., complementary slackness holds), we can clearly see: infinitesimal changes of $\hat{w}$ can not move $\nabla_{x}f(\hat{x})$ outside of $G(x),$ and hence can not change the solution $\hat{x}$. Therefore, Jacobian $\nabla_{\hat{w}}x^\ast(\hat{w})$ is zero. We would like to emphasize, that this result is *completely independent* of the function class of $f.$
>
> Importantly, the zero-gradient problem can occur even if the constraints are non-linear. For example, suppose $x\in\mathbb{R}^2$ and $\mathcal{C}$ is a convex lens -- an intersection of two disks. Then, $\mathcal{C}$ has two vertices where the gradient cone is two-dimensional. Hence, if the gradient of the internal objective $\nabla_xf(x, \hat{w})$ lies in the interior of either of these cones, the Jacobian $\nabla_{\hat{w}} x^\ast(\hat{w})$ is a zero matrix.
>
> In the experiments, we aim at demonstrating that the zero-gradient problem indeed occurs in practice. We consider the portfolio optimization problem an interesting benchmark as it allows us to smoothly vary its true objective between linear and quadratic regimes.
> Figures 3a and 4 of the main paper show that the performance of our $r-$smoothing approach is better than that of the standard algorithm. We believe that this happens due to the zero-gradient problem, but we also agree that the performance plots are not sufficient evidence. To support this point, we ran additional experiments. In the attached PDF file, Figure 2 shows the norm of the loss function gradient, $\|\nabla_{\theta}f(\hat{x}, w) \|_2$, and the number of active constraints during training. We compared the exact differentiation [2] of the QP approximation and our $r-$smoothing method for linear and quadratic versions ($\lambda=0$ and $\lambda=2$) of the portfolio optimization problem.
>
> These experiments demonstrate that with training, more constraints become active, and consequently, the gradient norm decreases. This process can be observed in both linear and quadratic cases, but is much more prominent in the former. As the QP internal problem is used in both cases, the difference can not be explained by the properties of the internal problem itself. Our explanation, also provided in lines 310-312 of the main paper, is that when the true objective $f$ is linear the true optimal solution lies on the boundary of the feasibility set $\mathcal{C}$. Because of that, the gradient of the loss function $\nabla_{x}f(\hat{x}, w)$ pushes the predicted solution $x^\ast(\hat{w})$ to the boundary of $\mathcal{C}$, and hence more constraints get activated. Then, based on Lemma 3.4, the null-space of the Jacobian $\nabla_{\hat{w}}x^\ast$ becomes larger, and hence the zero-gradient problem is more likely to occur.
>
> **2. The experiments have no baselines**
>
> To the best of our knowledge, differentiation through convex programs [2] is considered to be the ultimate approach for convex non-linear P\&O problems, as it computes the exact derivative (which we show can be non-informative). We are not aware of other works studying approximations for the P\&O loss in this case. However, it is true that there exist various methods for approximate differentiation of linear problems. In the attached PDF file, we compared our $r-$smoothing method against SPO+ surrogate loss [3] (labeled ''SPO+'' in the figure), mean-squared error $\|\hat{w}-w\|^2_2$ (''MSE''), and perturbation-based approach [4] (''perturbed''). The method labeled ‘’standard’’  corresponds to using the QP approximation and computing its exact derivative using the results from [2]. This method is equivalent to quadratic regularization from [1], and hence we did not include the latter in the baselines. As SPO+ and perturbation-based approaches are only applicable in the case of linear problems, we used the linear portfolio optimization problem ($\lambda=0$) and the OPF problem in these new experiments. The results demonstrate that our method performs better than the baselines.
>
> **3. The paper is poorly written**
>
> Thank you for pointing out this omission. We will describe our contribution properly in the abstract and introduction. We will also try to provide better explanations for our main results.
>
> [1] Bryan Wilder et al. Melding the Data-Decisions Pipeline: Decision-Focused Learning for Combinatorial Optimization (2019)\
> [2] Akshay Agrawal et al. Differentiable convex optimization layers (2019)\
> [3] Adam N Elmachtoub and Paul Grigas. Smart predict, then optimize (2017)\
> [4] Quentin Berthet et al. Learning with differentiable perturbed optimizers (2020)\

---

> > ### Comment · Reviewer_1CVQ · 2023-08-12
> > **Response to Rebuttal**
> >
> > Responding point-wise.
> >
> > ### Novelty of zero-gradient theorem
> > Despite the fact that nothing has ever been explicitly proved for the nonlinear case, I don't believe that the theorem is particularly novel for the following reasons:
> > 1. While I understand that the literature has typically talked about this issue in the context of linear programs, **the argument described in the paper's proof is exactly the same as that for the linear case**—(a) small perturbations to the parameters lead to the same decisions leading to zero gradients, and (b) this happens at the corners/edges of polytopes (or, generalizations thereof like the cylinder in Figure 1). For example, Figure 1 in Elmachtoub and Grigas (2017) and (to a lesser extent) Figure 1 in Berthet et. al. (2020) have the 2D equivalent of the Figure 1 in this paper. IMO, the paper's proof just frames this argument using the language of KKT conditions.
> > 2. **It is also understood in the literature that this is not because of the linearity of $f$**, but rather an interaction between the objective and the constraints. The only reason that this issue doesn't typically arise in QPs because (a) the optima for the quadratic term (typically the origin) is on the interior of the feasible region and (b) the gain from the quadratic term by moving away from the boundary outweighs the gain from the linear term in moving towards the boundary. If any of these conditions is broken, there can be zero-gradient issues, so I don't doubt that that zero-gradient issues arise in the experiments. For example, even in the simple 1D case, if we're trying to find the gradients of $x^*(\alpha) = \arg\min_x \alpha x^2 - 10x$  s.t.  $-1 \leq x \leq 1$ (which is a QP) with respect to $\alpha$, for values $0 < \alpha < 5$ we would have zero-gradients because the gain from moving away from the boundary is not high enough. The literature doesn't typically talk about general non-linear problems because there aren't many efficient ways to solve such problems even when the parameters are known.
> >
> > ### Literature related to r-smoothing-like surrogates
> > While it's true that the papers cited above are the most popular solutions for P&O, there has been a lot of recent work that is closer to the $r-$smoothing method that the paper proposes. For example, Sahoo et. al. [1] suggest using a surrogate gradient along with a projection component that seems very similar to the $x^*_{QP}$ surrogate proposed in the paper. The form of $x^*_{QP}$ also seems similar to recent E2E approaches (e.g., [2]) where the goal is to predict the decision directly and then use projection to ensure that it's feasible. It seems like the "decisions" such E2E models would learn is similar to the parameters $\hat{w}$ of $x^*_{QP}(\hat{w})$.
> >
> > [1] Sahoo, Subham Sekhar, et al. "Backpropagation through combinatorial algorithms: Identity with projection works." arXiv preprint arXiv:2205.15213 (2022).
> > [2] Cristian, Rares, et al. "End-to-End Learning for Optimization via Constraint-Enforcing Approximators." Proceedings of the AAAI Conference on Artificial Intelligence. Vol. 37. No. 6. 2023.
> >
> > ### Empirical Results
> > I also have a couple of questions about the new results:
> > 1. **Exact may not be the "ultimate approach":** As described above, there may be zero-gradients even for QPs if the quadratic term is too weak. This is why the idea from Wilder et. al. (2019), i.e., to *tune the L2 regularization as a hyperparameter* in order to have good performance, is different from the "exact" approach described in your experiments. It would be useful to know how well such a baseline does.
> > 1. **Linear Case:** SPO+ typically does quite well in the linear domain. Even MSE is provably optimal in the limit of infinite data and model capacity. They also don't run into the issue of having too many active constraints. However, there seems to be a big gap between the performance of your approach and these past approaches... Do you have any intuition for why $r-$smoothing is does so much better in your case?

---

> > > ### Author Response · Authors · 2023-08-13
> > >
> > > **Novelty of zero-gradient theorem.**
> > >
> > > This is true that our proof resembles the argument for the linear case. This is not surprising, as we generalize the known zero-gradient phenomenon onto a general convex optimization case. We do not think that this resemblance makes the results less important or novel. We agree, however, that our claim to discover the zero-gradient problem might cause misunderstanding. We will adjust the paper to avoid any over-claiming and so that the relation between our result and the prior works is clear.
> > >
> > > In short, our main theoretical result can be rephrased as *Number of active constraints defines the size of the null space of the Jacobian $\nabla_{\hat{w}}x^\ast(\hat{w})$. Moreover, normals to the active constraints are the basis of this null space. Hence, if the loss function gradient is contained in the span of these normals, the total gradient is zero.*\
> > > The known fact about linear problems is:
> > > *In the vertices of the constraint set, the Jacobian of the linear problem is a zero matrix*.
> > > Clearly, these two statements are not equivalent. We agree that deriving the former from the latter is not very complicated technically, but it is still an important result.
> > >
> > > The reviewer argues that our zero-gradient problem was, in fact, known before and that it does not really occur that often in practice. With all due respect, we disagree with both points and would like to explain our position. As an illustrative example, we will use the regularization method from Wilder et al. (2019) referenced by the reviewer. This method is designed to deal with the known version of the zero-gradient problem by adding $L_2$ regularization in the internal objective. However, as we explain below, it still suffers from the zero-gradient problem in 'our sense' (as in Theorem 3.9) both in theory and practice. We believe that this argument demonstrates that our zero-gradient problem indeed occurs in P\&O methods but it is not acknowledged anywhere.
> > >
> > > Let $f_\gamma (x, \hat{w})=\hat{w}^\top x - \gamma\|x\|^2_2.$ Then, we can rewrite it as $f_\gamma (x, \hat{w})= -\gamma\|x-\frac{\hat{w}}{2\gamma}\|^2_2$. As mentioned by the reviewer, by increasing $\gamma$ we can ensure that the $\arg\max_x f_\gamma(x, \hat{w})$ lies in the interior of the constraints set. However, it is **incorrect** that having such $\gamma$ resolves the zero-gradient problem. Suppose the true objective we maximize is linear, $f(x, w)=w^\top x$ (e.g., the bipartite matching experiment from Wilder et al.). Then, with training, the solution to the internal (regularized) problem, will move along $w$, until it reaches the boundary of $\mathcal{C}$. Then, it will move on the boundary and it might enter a non-smooth vertex and get stuck there. Hence, independently of the hyperparameter $\gamma$, the solution will get to the boundary (assuming 'good' data and neural network), if the loss function is linear. Hence, any method that differentiates through quadratic problems, even with tunable parameters, is, theoretically, suspectable to the zero-gradient problem.
> > > On the experimental side, the "exact" approach described in our experiments **is** equivalent to regularization from Wilder et al. (2019). As we show above, $f_\gamma (x, \hat{w})= -\gamma\|x-\frac{\hat{w}}{2\gamma}\|^2_2$. As the positive factor does not affect the optimization problem, using $f$ in the optimization problem is equivalent to using $-|x-\frac{\hat{w}}{2\gamma}\|^2_2$. Hence, if we scale the output of the neural network $\hat{w}$ by $\frac{1}{2\gamma}$, we see that QP approximation is equivalent to regularization.
> > > The scaling factor of the neural network output is a hyperparameter which we call $x_{scale},$  determine with grid search and report in Tables 3 and 4 of the paper. Therefore, there is a one-to-one correspondence between the quadratic regularization method and its hyperparameter $\gamma$ and our QP approximation with hyperparameter $x_{scale}$. In the experiments in the PDF attached to the rebuttal, you can see that the standard method (exact differentiation of QP) is indeed initialized with no active constraints. However, during training, the solution moves to the boundary and the zero-gradient problem occurs.
> > >
> > > To summarize, we would like to emphasize that, to the best of our knowledge, in the context of predict-and-optimize, all existing methods designed to deal with non-linear problems are based on computing the Jacobians of these problems. Our paper shows that such methods are likely to face the zero-gradient problem if the optimal solutions happen to be on the boundary and proposes a solution to this.

---

> > > > ### Comment · Reviewer_1CVQ · 2023-08-14
> > > >
> > > > I have some high-level comments, but first let us get on the same page about Wilder et al. (2019). I think that there's a difference in our understanding of their approach.
> > > >
> > > > >Then, with training, the solution to the internal (regularized) problem, will move along $w$, until it reaches the boundary of $\mathcal{C}$. Then, it will move on the boundary and it might enter a non-smooth vertex and get stuck there. Hence, independently of the hyperparameter $\gamma$, the solution will get to the boundary (assuming 'good' data and neural network), if the loss function is linear.
> > > >
> > > > In my understanding, at **train-time**, you always evaluate at $f_\gamma$ not $f$. Specifically, you are trying to maximize $f\_\gamma(x^*\_\gamma(\hat{w}), w)$ **not** $f(x^*\_\gamma(\hat{w}), w)$, where $x^*\_\gamma = \arg\max_x f\_\gamma(x, \hat{w})$. The minima of $f(x^*\_\gamma(\hat{w}))$ is at $\hat{w} = w$ and so, for reasonable initial values of $\hat{w}$ and high enough values of $\gamma$, you will never reach the boundary. At **test-time** you use your learned $\phi^* = \arg\max\_\phi f\_\gamma(x^*\_\gamma(\phi(o)), w)$ and solve the linear problem, i.e., $x^* = \arg\max\_x \phi^*(o)^T x$ in order to calculate the regret. Even if you're unconvinced by this, consider a related approach [1] in which there's an explicit log barrier term that prevents the solver from ever going to the boundary.
> > > >
> > > > Do you still believe that there's a zero-gradient issue with past work (i.e., [1] or the modified version of Wilder et. al. (2019))?
> > > >
> > > > [1] Mandi, Jayanta, and Tias Guns. "Interior point solving for lp-based prediction+ optimisation." Advances in Neural Information Processing Systems 33 (2020): 7272-7282.

---

> > > > > ### Author Response · Authors · 2023-08-14
> > > > >
> > > > > We thank the reviewer for this clarification and there is indeed a big difference between our understanding of the work of Wilder et al.
> > > > > If we understood the reviewer correctly, they say that during training, $f_\gamma$ is used not only as the *internal objective*, i.e., to compute $x_\gamma^\ast(\hat{w})=\arg\max f_\gamma(x, \hat{w})$, **but also** as the *true objective*, i.e., we use $\frac{\partial{f_\gamma(\hat{x}, w)}}{\partial{\hat{x}}}$ to do gradient update steps.
> > > > >
> > > > > We are very confused about this statement, as we have not seen any hints of that in Wilder et al. Moreover, we looked at the paper from Ferber et al. written by the same people that combines the regularization method with the cutting plane approach. There, Equations 3a-3c and the discussion thereof specify that the loss used for training is the *true* objective $f$ ($c^\top\hat{x}$ in their notation).
> > > > >
> > > > > We also find it counterintuitive: there seems to be no reason to optimize for $f_\gamma$ instead of true $f$ as both are perfectly differentiable. Besides that, for higher values of $\gamma$, the difference between the true objective $f$ and $f_\gamma$ becomes more different. Hence, optimizing for $f_\gamma$ makes even less sense.
> > > > >
> > > > > We believe that the method in Wilder et al. trains the predictor to optimize the true objective and the $L_2$ regularization is used to make $x^\ast()$ differentiable. Hence, we are confident in our previous argument regarding zero-gradients in Wilder et al.
> > > > >
> > > > >
> > > > >
> > > > > Aaron Ferber, Bryan Wilder MIPaaL: Mixed Integer Program as a Layer (2020)

---

> > > > > > ### Comment · Reviewer_1CVQ · 2023-08-14
> > > > > >
> > > > > > It's not necessarily counter-intuitive, imo. While $f_\gamma$ and $f$ do diverge in absolute value as $\gamma$ grows, the optima are the same, i.e., $\arg\max_{\hat{w}} f\_\gamma(x^*\_\gamma(\hat{w}), w) = \arg\max_{\hat{w}} f(x^*(\hat{w}), w) = w$. As a result, using it to train the predictive model $\phi$ will lead to optimal predictions.
> > > > > >
> > > > > > However, taking a step back, the claims being made is that past methods run into the zero-gradient issue. While I agree that the authors' interpretation of Wilder et. al. (2019) lead to zero-gradients, I don't think that this is true for past methods in general. Specifically, the modification that I described, as well as Mandi and Guns (2020), add regularization terms that ensure there will be no zero-gradients. Then, whether the $r-$smoothing approach is a "good surrogate" requires comparing to such past work, which the authors have addressed to some extent in the reviews.

---

> > > > > > > ### Author Response · Authors · 2023-08-14
> > > > > > >
> > > > > > > We agree that the interpretation suggested by the reviewer is an alternative way to deal with zero-gradients. The sole fact $\hat{w}=w$ minimizes the suggested loss function is insufficient to argue for this approach -- the same holds for e.g., MSE loss. We believe that P\&O is most important than the original $w$ can not be perfectly predicted and we inevitably have some errors. In this case, it is important to minimize the error using correct loss (e.g., regret as opposed to MSE).
> > > > > > >
> > > > > > > This interpretation is not an established method (even though it can become one), as it has never been studied/considered in the literature (as we argue above, Wilder et al. use *true* objective as a loss function). Hence, it was impossible for us to even consider comparing to this method.
> > > > > > >
> > > > > > > The method of Wilder et al. runs into zero-gradient problems and there might be multiple ways to deal with that (the idea suggested by the reviewer might be one of them). We agree that the method from Mundi and Guns (2020) enforces the decisions to be in the interior of the constraints set. This ensures no zero-gradients, but, on the other hand, limits the representational capability of the model.
> > > > > > >
> > > > > > > **Update** We confused the reference for Mundi ang Guns for another work, our initial comment is removed.

---

> > > > ### Comment · Reviewer_1CVQ · 2023-08-14
> > > >
> > > > > In short, our main theoretical result can be rephrased as Number of active constraints defines the size of the null space of the Jacobian. Moreover, normals to the active constraints are the basis of this null space. Hence, if the loss function gradient is contained in the span of these normals, the total gradient is zero. The known fact about linear problems is: In the vertices of the constraint set, the Jacobian of the linear problem is a zero matrix. Clearly, these two statements are not equivalent.
> > > >
> > > > Here's another question; it seems like the argument about what separates this theorem from the linear case (where the Jacobian is exactly zero) is the claim that the gradient may sometimes be in the null space of the normals even when the Jacobian matrix isn't exactly zero. However, will this ever be the case? It doesn't seem like it because, if that were true, such a Jacobian would have zero partial derivatives with respect to variables not contained in this null space. That, in turn, means that these variables do not affect the objective and are also not part of the active constraints (so you could safely remove the variable from the optimization problem and have the Jacobian be exactly zero)? I can't see why this null space characterization is strictly more expressive than the all-zeros characterization (and hence adds value).

---

> > > ### Author Response · Authors · 2023-08-13
> > >
> > > **Literature related to r-smoothing-like surrogates**
> > > We thank the reviewer for providing these references, we were not aware of them before the rebuttal. We agree that the method [1] is of the same nature as our $r-$smoothing and we will include it in the discussion and related work. It is worth mentioning, however, that [1] focuses on the linear case and it is not immediately clear how it can be extended to non-linear case.
> > > The paper [2] operates in a slightly different setting, as its goal is *to solve* a given optimization problem in a differentiable way.  The authors present their method as a neural approximate differentiable solver. In the experiments, they show that they can train it to solve linear problems with good accuracy. It is not clear, however, how this result compares to the existing P^&O methods. Moreover, their method is probably also susceptible to the zero-gradient problem -- if the neural network indeed learns to solve the optimization problems, it will have zero-gradients when the solution is on a vertex of the polytope.
> > >
> > > **Empirical Results**
> > > As SPO+ loss [3] and perturbed optimizers method [4] derive approximate ways to differentiate linear problems using different approaches, it is not an easy task to perform a theoretical comparison of them with our $r-$smoothing method. Our intuition regarding the differences in methods is that it comes from different internal problems. For $r-$smoothing, we use QP internal problem, while the two other methods use linear internal problems. Hence, they have a simpler model class for the $x^\ast(\hat{w})$ as it can only take values in the vertices of $\mathcal{C}$.
> > >
> > > [1] Sahoo, Subham Sekhar, et al. "Backpropagation through combinatorial algorithms: Identity with projection works." arXiv preprint arXiv:2205.15213 (2022).\
> > > [2] Cristian, Rares, et al. "End-to-End Learning for Optimization via Constraint-Enforcing Approximators." Proceedings of the AAAI Conference on Artificial Intelligence. Vol. 37. No. 6. 2023.
> > > [3] Adam N Elmachtoub and Paul Grigas. Smart predict, then optimize (2017)\
> > > [4] Quentin Berthet et al. Learning with differentiable perturbed optimizers (2020)

---

### Official Review · Reviewer_Yhee · 2023-07-05

**Soundness:** 3 good
**Presentation:** 4 excellent
**Contribution:** 3 good
**Rating:** 6
**Confidence:** 3

**Summary:**

This paper identifies the zero-gradient problem in Predict and Optimize (P&O) for convex optimization and proposes a method to address it. The method is based on using a Quadratic Programming (QP) approximation for computing decisions, smoothing the feasibility region around the current solution to reduce the dimensionality of the null space to one, and adding a projection distance regularization term. The proposed method demonstrates significant improvements for convex P&O problems with many constraints and with the true optimum lying on the boundary of the feasibility set.

**Strengths:**

Originality: The paper identifies a previously unnoticed problem in convex optimization and proposes a novel method to solve it.

Quality: The proposed method is technically sound, and the experiments demonstrate its effectiveness in addressing the zero-gradient problem.

Clarity: The paper is well-written and clearly explains the concepts and methodology.

Significance: The proposed method has the potential to improve optimization in convex P&O problems, which are common in various domains.

**Weaknesses:**

Insufficient experiments: The paper might lack a comprehensive set of experiments or fail to compare the proposed method with alternative approaches. This could make it difficult for readers to evaluate the true effectiveness and novelty of the proposed method.

**Questions:**

In Section 3.2, the authors present a Quadratic Programming (QP) approximation for computing decisions, which substitutes the original objective function f(x, w) with an alternative objective function $f_{QP}(x, w)$. Although it may appear counterintuitive that using a separate objective function does not influence the final solution, the key motivation behind the QP approximation is its ability to simplify Jacobian computations and tackle the zero-gradient problem. However, in the experimental section, the authors only demonstrate the effectiveness of the QP approximation in one case, varying the parameter $\lambda$ and presenting the results in Table 1. Consequently, I am curious about how this approximation performs in other optimization problems. Additionally, the authors only test their overall method in two cases, which may not sufficiently demonstrate the method's robustness and generalizability. I would appreciate further insights into the performance of the proposed approach across a wider range of problems and scenarios.

Is the zero-gradient problem universal in all predict and optimize for convex problems?  For example, if the surrogate function is not KKT-based and instead uses an extra large penalty term to penalize the constraints, would the same phenomenon exist[1], or is it specific to KKT-based techniques or the standard technique mentioned in the experiments? It would be helpful if the authors could clarify the scope of their contributions and avoid overclaiming.

In Line 260, the authors introduce the parameter $\alpha$, but it seems that this parameter is not discussed in the experiments section. Can the authors provide more information on how $\alpha$ is chosen or tuned in the experiments, and how its choice affects the performance of the proposed method?

Does the term "standard" in the figure refer to the work of [2]? It appears that the authors do not explicitly mention this term in the main text. Can the authors clarify the connection between the "standard" and the cited work, and if possible, provide a clearer definition or explanation of the term within the paper?


References:

[1] A Surrogate Objective Framework for Prediction+ Programming with Soft Constraints. Advances in Neural Information Processing Systems, 2021.

[2] Differentiable convex optimization layers. Advances in Neural Information Processing Systems, 2019



**Limitations:**

See Weaknesses

---

> ### Author Rebuttal · Authors · 2023-08-07
>
> Thank you very much for reviewing our work.
> To the best of our knowledge, differentiation through convex programs [2] is considered to be the ultimate solution for convex non-linear P\&O problems, as it computes the true gradient. We are not aware of other works studying approximations for the P\&O loss in this case. However, it is true that there exist various methods for approximate differentiation of linear problems. In the attached PDF file, we compared our $r-$smoothing method against SPO+ surrogate loss [3] (labeled ''SPO+'' in the figure), mean-squared error $\|\hat{w}-w\|^2_2$ (''MSE''), and perturbation-based approach [4] (''perturbed''). As SPO+ and perturbation-based approaches are only applicable in the case of linear problems, we used the linear portfolio optimization problem ($\lambda=0$) and the OPF problem in these new experiments. The results demonstrate that our method performs better than the baselines.
>
> 1. We fully agree, that it would be beneficial to test QP approximation on a broader spectrum of problems. However, we could not find any benchmark problems for P\&O with convex, nonquadratic objectives. Instead, we ran an additional experiment with a modified portfolio optimization problem. We substituted the linear term in the objective with the LogSumExp:
> $$f(x, w, Q)= \log(\sum_i e^{w_ix_i}) - x^\top Q x.$$
> This problem does not necessarily makes a lot of practical sense, but it allows us to test how well the QP approximation works when the true objective $f$ is a convex, non-quadratic function. In Figure 3 in the attached PDF, we compare the QP approximation without (labeled 'QP') and with (labeled '$r-$smoothing') our $r-$ smoothing technique to using the true function $f$ in the internal problem (labeled 'true $f$'). The results demonstrate that QP approximation both with and without smoothing performs better than using the true $f$. We will run more experiments comparing QP approximation with non-quadratic internal problems and include them in the appendix of the paper.
>
> 2. The zero-gradient problem is a property of convex constrained optimization problems. Essentially, in some regions, the solution mapping $x^\ast(\hat{w})$ might be constant in certain (or all) directions. Hence, our results affect those methods that are based on differentiating $x^\ast(\hat{w})$. The method from the reference [1] operates in a different regime: it softens the constraints by adding them to the objective function. In this case, $x^\ast(\hat{w})$ becomes an unconstrained $\arg\max$. As there are no constraints, our results are not applicable here. However, the softened loss function (Eq. 6 in [1]) is non-convex and might have an arbitrarily complex landscape. Studying whether this landscape has flat regions would be an interesting task. We will also adjust the paper to make it clear, what types of methods are affected by our results.
>
> 3. $\alpha$ is a hyperparameter that defines the weight of the projection distance regularization in the loss function, i.e., it determines how strongly $\hat{w}$ is pulled towards $\mathcal{C}.$ For each experiment (OPF problem; portfolio optimization problem with different values of $\lambda$), we determine the best value of $\alpha$ by running a grid search. Search spaces and final values are reported in the supplementary material, in Tables 1-4.
>
> 4. By ``standard’’, we indeed mean the exact method to compute the Jacobian $\nabla_{\hat{w}} x^\ast$ introduced in [2]. We will emphasize this more in the experiment's description.
>
> [1] Kai Yan et al. A Surrogate Objective Framework for Prediction+ Programming with Soft Constraints.  (2021)\
> [2] Akshay Agrawal et al. Differentiable convex optimization layers (2019)\
> [3] Adam N Elmachtoub and Paul Grigas. Smart predict, then optimize (2017)\
> [4] Quentin Berthet et al. Learning with differentiable perturbed optimizers (2020)\

---

> > ### Comment · Reviewer_Yhee · 2023-08-16
> >
> > I still have some concerns regarding the scope and claims of this paper. Although the authors have clearly stated that they are addressing the Zero-Gradient Problem in Predict and Optimize for Convex Optimization, I believe that this zero-gradient issue does not necessarily need to be discussed in every predict and optimize framework, as exemplified at least by the previous work I cited, [1].

---

### Official Review · Reviewer_8mJH · 2023-07-09

**Soundness:** 3 good
**Presentation:** 3 good
**Contribution:** 3 good
**Rating:** 6
**Confidence:** 1

**Summary:**

This paper studies predict and optimize problem which utilizes machine learning to predict unknown parameters of optimization problems. The paper identifies the zero-gradient problem and proposes a method to solve this issue. Additionally, the paper conducts an experimental study to verify the proposed method.

**Strengths:**

1. This paper is technically sound. The claims regarding the zero-gradient problem in the paper are well-supported by theoretical analysis. The assumptions are clearly presented, and proof ideas are discussed after each theorem. The efficiency of the proposed solution to the zero-gradient problem is verified by the experimental results.
2. The paper is well-organized. It begins by introducing the problem formulation of predict and optimize and discusses the typical methods used to solve the problem. Then, the paper introduces the zero-gradient problem along with the theoretical analysis. Finally, the proposed solution and experimental results are presented.

**Weaknesses:**

The paper can be improved by also experimentally demonstrating the yet noticed zero-gradient problem claimed in his paper.  Demonstrating the the consistency between the theoretical findings and experimental observations can enhance the significance of this paper.

**Questions:**

1. Line 90-91. The parameter w is unknown. In this case, how can one minimize the loss function define in Eq (1)?
2. Is there any way to theoretically analyze and show how the proposed OP and local smoothing methods solve the zero-gradient problem?

---

> ### Author Rebuttal · Authors · 2023-08-07
>
> Thank you for taking the time to evaluate our work!
>
> In the paper, Figures 3a and 4 show that the performance of the $r-$smoothing approach is better than that of the standard algorithm. We believe that this happens due to the zero-gradient problem, but we also agree that the performance plots are not sufficient evidence. To demonstrate that the zero-gradient problem really occurs, we ran additional experiments. In the attached PDF file, Figure 2 shows the norm of the loss function gradient, $\|\nabla_{\theta}f(\hat{x}, w) \|_2$, and the number of active constraints during training. We compare the standard approach and our $r-$smoothing method for linear and quadratic versions ($\lambda=0$ and $\lambda=2$) of the portfolio optimization problem. The results confirm -- in the standard method, more constraints become active with training and the gradient norm decreases. This phenomenon is more prominent when the true objective is linear. Our explanation for that is given in lines 310-312 of the paper:\
> *... linear true objective pushes the decision $\hat{x}$ towards the boundary of $\mathcal{C}$, and hence it is more likely to enter points with a large gradient cone. For the more quadratic objectives, the true maximum is often in the interior of $\mathcal{C}$ and hence the zero-gradient problem occurs less often.*
>
> 1. It is assumed that the parameter $w$ is unknown at the moment when the decision is made, but it is accessible during training, e.g., to evaluate the loss function. It is a common assumption for a supervised learning setup.
> 2. In Theorem 3.5 we demonstrate that the zero-gradient problem occurs when the gradient of the true objective $\nabla_{x}f(\hat{x},w)$ lies in the null-space of the Jacobian $\nabla_{\hat{w}}x^\ast.$  In Lemma 3.4, we show that the dimensionality of the null space, in turn, is defined by the number of active constraints, i.e., the more constraints are active the more dimensions the null space contains. The $r-$ smoothing method introduced in Section 3.3 is based on approximating the Jacobian, $\nabla_{\hat{w}}x^\ast(\hat{w})\approx \nabla_{\hat{w}}x_r^\ast(\hat{w}),$ such that null-space of this approximation is always zero- (if no constraints are active) or one-dimensional (Property 3.8). Therefore, by design, the $r-$smoothing method will encounter the zero-gradient problem only when the true gradient $f_x(\cdot, w)$ aligns perfectly with the one-dimensional null space of $\nabla_{\hat{w}}x_r^\ast(\hat{w})$. To deal with this, we introduce the projection distance regularization in Eq. 6.
> It is worth mentioning that QP approximation on its own does not affect the zero-gradient problem. However, the Jacobian $\nabla_{\hat{w}}x^\ast$ for the QP approximation is simple to analyze. We exploit this in the proof of Theorem 3.9, which shows that computing gradient steps using $r-$smoothing combined with the QP approximation is guaranteed to at least not decrease the performance.

---

### Official Review · Reviewer_ZB3i · 2023-07-11

**Soundness:** 3 good
**Presentation:** 2 fair
**Contribution:** 2 fair
**Rating:** 4
**Confidence:** 4

**Summary:**

Predict+Optimize (P+O) is an emerging paradigm that lies in the intersection of classical optimization  and machine learning. Specifically, it considers the setting where a parameterized optimization problem:

$$x^{\star}(w) = \operatorname*{argmin}_{x} f(x,w) \text{ subject to } x \in \mathcal{C}$$

must be solved yet the parameters $w$ are unknown. Given observational data $o$ that is correlated with $w$, a natural approach is to train a machine learning model $\hat{w} = \phi_{\theta}(0)$ so that $\hat{w} \approx w$. Then at test time, we solve

$$ x^{\star}(\hat{w}) = \operatorname*{argmin}_{x} f(x,\hat{w}) \text{ subject to } x \in \mathcal{C} $$

The secret sauce to P+O is, instead of training to minimize prediction error $\|\hat{w} - w\|^2$, to use a loss function aligned with the actual goal {\em i.e.} that encourages $ x^{\star}(\hat{w}) \approx  x^{\star}(w)$. There are several ways to do this, but one is to simply reuse the objective function and train $\phi_{\theta}$ so as to minimize

$$ \mathbb{E}_{(o,w)}\left[f(x^{\star}(\hat{w}, w)\right]  $$

where $\hat{w} = \phi_{\theta}(o)$. Although a formula for the derivative of this loss is well-known, the core claim of this paper is that this derivative is less informative than previously thought. In fact, it is frequently zero. This idea is formalized through a theorem. The authors then propose a way to overcome this aplty named "zero-gradient" problem. Finally, the paper is rounded out by numerical experiments on two datasets.


**Strengths:**

- The main strength of this paper is Lemma 3.4 and Theorem 3.5, which crystalize the core claim of this paper. This result is surprising, but I checked the proof to the best of my ability and I believe it is correct. This result is an important reality check for the field of Predict-and-Optimize.
- I enjoyed reading the proofs. The result of [1] were new to me. I liked the way the strict complementary slackness is used in the proof of Lemma 3.4
- Adding to the above, the authors do a good job of making their core results accessible through intuitive explanations and diagrams.

[1] Anthony V Fiacco. _Sensitivity analysis for nonlinear programming using penalty methods_ (1976).

**Weaknesses:**

- Reusing the parameterized objective function $f(\cdot,w)$ as the loss function for training is not the only way to do P+O. One could also use the SPO+ loss [1], a perturbation based approach [2], or the least squares loss $\|x^{\star}(w) - x^{\star}(\hat{w})\|^2$ [3]. This should be mentioned as the zero-gradient theorem need not apply in these settings.
- I am perplexed at the stated motivation behind the quadratic programming approximation. While it is true that $f_{QP}(x,\hat{w}) = \|x - \hat{w}\|^2$ is strongly concave, and so on, it need not bear any relation to the actual problem we wish to solve, namely $f(x,w)$. So, this seems to run counter the spirit of P+O. The only case that makes sense to me is when $f(x,w) = w^{\top}x$. Expanding out we get:

$$ f_{QP}(x,\hat{w}) = \|x - \hat{w}\|^2 = -2\hat{w}^{\top}x + \|x\|^2 + \|\hat{w}\|^2 $$

So ignoring the irrelevant $\|\hat{w}\|^2$ term,  it appears the authors are simply proposing to add a quadratic regularizer, which has been explored thoroughly in the literature (see [4] and elsewhere).  Could the authors comment on this?
- I find the motivation behind $r$-smoothing a little opaque too. It seems as though the solution to the $r$-smoothed problem $P_{r}(\hat{x}\hat{w})$ might not be feasible (i.e. might not lie in $\mathcal{C}$). Is this correct?
- Using the Jacobian $\nabla_{\hat{w}} x_r^{\star}(\hat{w})$ in place of $\nabla_{\hat{w}}x^{\star}(\hat{w})$ is, as you show, essentially the same as just replacing $\nabla_{\hat{w}}x^{\star}(\hat{w})$ with the identity (independent of what $r$ is). This procedure is already well-studied, see [3, 5--7]. These papers should be cited and discussed.

*Minor Stuff:*
- In Figure 2, the smoothed feasibility region $\mathcal{C}_r(\hat{x}\hat{w})$ is the disk (i.e. the interior of the circle) right? If yes, this should be made clear in the caption and figure. Right now it looks as though the feasible region is just the boundary.
 - In Definition 3.7, as the scale of $r$ doesn't really matter, I'd recommend not normalizing and simply writing $c = \hat{x} - r\nabla_xf(\hat{x},\hat{w})$.
 - The experiments in Section 4 feel like ablation studies (i.e. just removing one element at a time from your proposed approach). I would like to see some benchmarking results, e.g. comparing the performance of your proposed algorithm to existing P+O approaches. You may find the benchmarking software PyEOPO useful for this [8]

[1] Adam N Elmachtoub and Paul Grigas. _Smart predict, then optimize_ (2017)

[2] Quentin Berthet et al. _Learning with differentiable perturbed optimizers_ (2020)

[3] Daniel McKenzie et al _Faster predict-and-optimize with three-operator splitting_ (2023)

[4] Bryan Wilder et al _Melding the Data-Decisions pipeline: Decision Focused learning for combinatorial optimization_ (2019)

[5] Samy Wu Fung et al _JFB: Jacobian-Free Backpropagation for Implicit Networks_ (2022)

[6] Zhengyang Geng et al _Is attention better than matrix decomposition?_ (2022)

[7] SS Sahoo et al _Backpropagation through combinatorial algorithms: Identity with projection works_ (2022)

[8] Tang and Khalil _PyEPO: A PyTorch-based end-to-end predict-then-optimize library for linear and integer programming_ (2023).

**Questions:**

See "weaknesse" above.

**Limitations:**

N/A.

---

> ### Author Rebuttal · Authors · 2023-08-09
>
> *1. Insufficient benchmarks*\
> Thank you for providing us with these references. We have not included more benchmarks in the submission since all the papers we are aware of focus on linear/combinatorial problems. The reason behind that is simple -- the zero-gradient problem was not noticed before, and hence the exact differential optimization method from [1] was considered to be the ultimate solution to convex nonlinear P\&O problems.
> There indeed exist various methods to approximately differentiate linear problems. In the attached PDF, we compare our $r-$smoothing method against SPO+ loss [2] (labeled ''SPO+'' in the figure), mean-squared error $\|\hat{w}-w\|^2_2$ (''MSE''), and perturbation-based approach [3] (''perturbed''). We have not included the argmax loss [4] $\|x^\ast(w) - x^\ast(\hat{w}) \|_2^2,$ as it is also susceptible to the zero-gradient problem (since it includes $x^\ast(\hat{w})$) but optimizes a surrogate loss.
> As SPO+ and perturbation-based approaches are only applicable to linear problems, we used the linear portfolio optimization problem and the OPF problem in these new experiments. The results indicate that our local $r-$smoothing method outperforms all these benchmarks.
>
> *2. QP approximation*\
> We agree that the motivation for using QP approximation independently of the true objective $f$ might be unclear, so we elaborate on it here. In our eyes, P\&O is mostly about enforcing constraints on the output of predictive models, e.g., neural networks. Indeed, if the constraints set $\mathcal{C}$ is simple, e.g., a hypercube $\\{x\in\mathbb{R}^n| 0 \leq x \leq 1 \\}$, we do not need any of the P\&O methods. Instead, we can simply use an activation function that constrains the output (e.g., sigmoid) and then train the neural network to predict $\hat{x}$ by performing gradient descent on $f$. However, when constraints are more complex, this approach falls apart and we need a differentiable constrained optimization layer -- and this is exactly what P\&O provides us.
> The key motivation behind P\&O [2] is that we do not need the internal objective $f(\cdot,\hat{w})$ to be similar to the true objective $f$ -- we only want it to yield a good decision $\hat{x}$. From this perspective, QP approximation seems like a logical next step, as it is the simplest constrained optimization layer that can output any point in $\mathcal{C}$.
> When using QP approximation, we leave the heavy lifting related to ‘understanding’ the dependency between the features $o,$ the true objective $f$, and the optimal solution $x^\ast(w)$ to the predictor, while the P\&O module is used to enforce the constraints in a differentiable way.
>
> *3. Smoothing*\
> We would like to share our intuition behind the $r-$smoothing method. In Section 3, we show that the zero-gradient problem arises in the vertices where the constraints set $\mathcal{C}$ is not smooth, i.e., multiple constraints are active. If we could smooth all such vertices, it would resolve the zero-gradient problem almost entirely (the null space of the Jacobian can still be one-dimensional when the optimal solution is in the interior of $\mathcal{C}$). In fact, it is known (see e.g., [5]), that any convex polytope can be approximated by a smooth convex set with arbitrarily good accuracy. Suppose that we can use such a smooth approximation instead of $\mathcal{C}$. Then, the solution of the resulting problem can be made arbitrarily close to the true optimal solution, and yet the zero-gradient problem will be almost gone.
>
> Taking this argument one step further, we can see that we do not really need to make the whole set $\mathcal{C}$ smooth. In fact, at every gradient step, we want to know *what would the Jacobian of the smoothed (globally) problem look like for current prediction $\hat{w}$*. The local $r-$smoothing method is designed to answer exactly that question.
>
> Importantly, the solution to the $r-$smoothed problem is defined such that its solution equals to the solution of the non-smoothed internal problem (see lines 224-225 of the paper). In fact, we do not need solve the smoothed problem because of that reason. As we use the QP approximation, the Jacobian $\nabla_{\hat{w}} x_r^\ast(\hat{w})$ can be also computed explicitly, without differentiating the KKT conditions. In theory, we expect that it is possible to use other internal problems (e.g., with the original $f$) instead of QP approximation, and then compute the Jacobian $\nabla_{\hat{w}} x_r^\ast(\hat{w})$ by differentiating the KKT conditions of the $r-$smoothed version of this problem. However, unlike with QP approximation, we don't have a proof for non-decrease (Theorem 3.9) in this case.
>
> *4.Similarities to Jacobian-free Backpropagation (JFB)*\
> Thank you very much for providing these references, we were not aware of them. Indeed, our approach seems to be similar in spirit to the idea of JFB. Specifically, references [4], [6] have a lot in common with our $r-$smoothing approach. However, we also see some differences. These works focus on the linear case: [4] requires linear constraints and [6] needs a linear objective.
> We will look deeper into JFB to better understand connections to our work and extend the related work section.
>
> *5.Response to the Minor stuff.*
> - You are correct about Figure 2; we will adjust it accordingly. Thank you for pointing that out!
> - Similarly, we agree with the remark about Definition 3.7.
> - We addressed this in the new experiments as described in the first paragraph.
>
> [1] Akshay Agrawal et al. Differentiable convex optimization layers (2019)\
> [2] Adam N Elmachtoub and Paul Grigas. Smart predict, then optimize (2017)\
> [3] Quentin Berthet et al. Learning with differentiable perturbed optimizers (2020)\
> [4] Daniel McKenzie et al. Faster Predict-and-Optimize with Davis-Yin Splitting (2023)\
> [5] Mohammad Ghomi Optimal Smoothing for Convex Polytopes (2004)\
> [6] SS Sahoo et al Backpropagation through combinatorial algorithms: Identity with projection works (2022)

---

> > ### Comment · Reviewer_ZB3i · 2023-08-13
> > **Response to Rebuttal**
> >
> > Thanks to the reviewers for addressing my comments so thoroughly! I will respond point-by-point.
> >
> > 1. Good job for implementing so many additional benchmarks in such a short time! In addition to testing other methods on the problem you introduce, I think it is crucial to test your method on problems already in the literature. The PyEPO library mentioned above makes this pretty easy.
> > 2. Unfortunately this motivation behind the QP approximation makes even less sense to me. I think P&O is significantly more general than enforcing the output of neural network to lie in a polytope $\mathcal{C}$. I maintain that the authors have essentially rediscovered the well-known trick of adding a quadratic regularizer to a linear objective introduced by Wilder et al. (I see you have experimented on a nonlinear objective described in the global rebuttal, but I don't understand how your QP approximation can achieve lower regret than using the true objective function)
> > 3. Thanks for clarifying the use of $r$-smoothing. I agree that it would be interesting to see how it works in conjunction with other internal problems.
> > 4. I agree that clarifying the relationship between your work and JFB (also known as one-step differentiation) is crucial.
> >
> > Since posting my review, I have thought more about this paper. It seems to me that the key ingredient to proving the zero gradient theorem is strict complementary slackness. I believe this holds for all linear programs, and generically for semi-definite programs. What about more general constrained optimization problems. For example, does strict complementary slackness always hold for quadratic programs?

---

> > > ### Author Response · Authors · 2023-08-14
> > >
> > > 1. We agree that using more of the already existing test problems is important to strengthen our paper. The initial reason why we have not used problems from PyEPO is that they are from the combinatorial optimization domain and are used to evaluate algorithms for linear/combinatorial P\&O methods. We will however extend our experiments with some of these problems to demonstrate the effectiveness of our method in this case.
> > > 2. We agree that the QP approximation we use is not novel and we do not claim it to be so. We treat it as a tool which, combined with $r-$smoothing, allows us to solve the zero-gradient problem both theoretically and experimentally. Besides, it offers computional benefit as its Jacobian can be computed analytically, without inverting the Hessian of the internal objective.
> > > As for experiments with LogExpSum objective, we believe the explanation is as follows: the LogExpSum objective has $n + n^2$ parameters (corresponding to the weghts $w$ and positive definite matrix $Q$). Hence, it results in a much more challeging problem for the neural network. This is generally true, as QP approximationg has the minimum possible number of parameters (equals to number of decision variables $n$).
> > > However, we also admit that we ran this new experiment in a really short time, and we could have not found the optimal hyperparamters for the method that uses the true objective. We will rerun this experiment more thoroughly and report the results in the paper.
> > >
> > > **Strict complementary slackness.**
> > >
> > > Strict complementary slackness does not *always* hold, even in the case of linear problems. For example, consider a two-dimensional square $[0,1]^2$ as the constraints set and let $f_{lin}(x,w)=w_1x_1 +w_2x_2$. In this case, the optimal solution $x^\ast$ is not unique but we can still use it as an example.
> > > Consider the prediction $\hat{w}=(0, 1)$ and decision $\hat{x}=(0, 1)$, where two constraints are active. Let $n_1=(0, 1)$ and $n_2=(-1, 0)$ be normals of these constraints. This point $\hat{x}$ is an optimal solution, but strict complementary slackness is violated -- $\nabla_x f_{lin}(x,\hat{w})=n_1 + 0n_2.$ Geometrically, it corresponds to the objective function gradient being on the *boundary* of the gradient cone at $x.$ In fact, as shown in Lemma 3.3 rephrasing known results from [1], $x^\ast$ is *non-differentiable* when strict complementary slackness is not satisfied. We can also see it geometrically -- slightly rotating the gradient of $f$ anti-clockwise ($w_1\downarrow$) will not change the solution while rotating it clockwise ($w_1\uparrow$) will make it jump.
> > >
> > > The same example holds for QP, e.g., consider an objective function $f_{qp}(x, \hat{w})=  \|x-\hat{w}\|^2_2$ and let $\hat{w}=(0, 5)$.
> > > In this case, the gradient at the optimal solution $\hat{x}=(0, 1)$ is also pointing up from $\hat{x}$. The same argument as for the linear case applies. In the QP case, however, rotating the gradient clockwise will not make the solution 'jump' but it will smoothly move it along the edge. Hence, in this case, $x^\ast$ has directional derivatives.
> > >
> > > In conclusion, we would like to say that the reviewer is correct that strict complementary slackness is crucial for our results, as when it is violated, Jacobian $\nabla_\hat{w} x^\ast$ is undefined. We would like to emphasize, however, that the points $\hat{w}$ that violate strict complementary slackness form a set of measure zero (as it requires landing the gradient exactly on the border of gradient cones) and hence can be neglected in practice.
> > >
> > > [1] Anthony V Fiacco. Sensitivity analysis for nonlinear programming using penalty methods (1976).

---

### Official Review · Reviewer_46Fp · 2023-07-31

**Soundness:** 4 excellent
**Presentation:** 3 good
**Contribution:** 3 good
**Rating:** 7
**Confidence:** 4

**Summary:**

This paper focus on the topic of "predict and optimize" and identify the zero-gradient problem. This issue is characterized by a situation where the gradient related to the best decision concerning parameters in machine learning models might be zero. This can occur even when assuming convexity, smoothness, and strict complementary slackness. The authors introduce a QP approximation and an r-smooth technique to address this problem, effectively reducing the likelihood of encountering the zero-gradient issue. The numerical results show the merits of their proposed approach.

**Strengths:**

1. Section 3.1 uses straightforward theoretical outcomes to shed light on a significant practical problem. This section is clear and insightful.
2. Section 3.3 stands out for its clarity and insight as well, particularly Theorem 3.9. Initially, I was skeptical that the gradient with local smoothness would accurately approximate the original gradient, but Theorem 3.9 effectively argued this point. Although the locally smoothed gradient may not ensure the fastest improvement in function value as the original gradient does, it can be guaranteed to at least be a non-decreasing direction.
3. Figures (b) and (c) look interesting, showing that the new methods outperform standard approaches during the final training phase. This aligns with the theories in Section 3.1, where, towards the end of training, the conditions described in Theorem 3.5 become more likely. This leads to training difficulties with the standard method, whereas the proposed techniques can overcome them. Further validation on this matter is needed, as indicated in the third point under "Weaknesses."

**Weaknesses:**

1. Sec 3.4, Algorithm 1. The notation seems unclear. What's the dimension of $f_x$ and $\hat{f_x}$? Why $f_x$ can be directly multiplied with $\nabla_{\hat{w}}x^*_r(\hat{w})$? Does it mean inner product of two vectors? What's the meaning of $f_x - f^0$ given $f^0$ is a scalar while $f_x$ is a gradient (vector)?

2. Sec 4.1, equation (8). What's the meaning of $w$? Does $f(x,w)$ mean the original function defined in (7) or the QP approximation defined in (9)? This is quite critical: if (8) measures the regret based on QP approximation rather than the original function, the experiment results would be meaningless. While if (8) measures the original function, the numerical results will be good.

3. Is it possible to provide the norms of the gradients you observed in the experiments? If the gradients calculated with your proposed approaches have larger norms than the traditional calculation way, it would be a more direct and strong support of your approach.

**Questions:**

see "Weaknesses"

**Limitations:**

see "Weaknesses"

---

> ### Author Rebuttal · Authors · 2023-08-07
>
> Thank you for identifying the strengths of our work as well as highlighting some important drawbacks.
>
> 1. There is indeed a typo in Algorithm 1, thank you for pointing that out. As $f_x, \hat{f}_x$ are the gradients of the objective function, they are vectors of size $n.$ Then, the definition of $f^0$ contains a typo -- it should read as $f^0:=\hat{f}_x\frac{f_x^\top \hat{f}_x}{\hat{f}_x^\top \hat{f}_x}$. Hence, $f^0$ is also an $n-$dimensional vector -- the orthogonal projection of $f_x$ onto $\hat{f}_x.$ Then, the difference $f_x - f^0 \in \mathbf{R}^n$ is the component of $f_x$ orthogonal to $\hat{f}_x$.
>
> 2. We agree that Eq. 8 is not properly presented. Also, there is a typo: the medium bracket should end after $\phi$, not $w$. Eq. 8 defines regret for the general case (notation from Sections 3.1-3.3). It uses the true objective function $f$, and the solution method affects it only via $x^\ast(\hat{w})$. In the case of the portfolio optimization problem, the unknown parameters are defined as $w=(p, Q)$.
>
> 3. Thank you for this suggestion. In the attached PDF file, Figure 2 shows the loss function gradient norm, $\|\nabla_{\theta}f(\hat{x}, w) \|_2$, and the number of active constraints during training. We compare the standard approach and our $r-$smoothing method for linear ($\lambda=0$) and quadratic ($\lambda=2$) versions of the portfolio optimization problem. The results confirm that the standard method suffers from the zero-gradient problem, especially when the true objective is linear. This provides another evidence to the explanation given on lines 310-312 of the paper:\
> ``... linear true objective pushes the decision $\hat{x}$ towards the boundary of $\mathcal{C},$ and hence it is more likely to enter points with a large gradient cone.''

---

### Author Rebuttal · Authors · 2023-08-09

We thank the reviewers for providing us with valuable feedback. To address the comments related to benchmarks and experiments, we conducted some additional experiments. We provide detailed responses individually for each reviewer, and in this text, we describe the new Figures that can be found in the attached PDF file.

**Figure 1.** To respond to fair criticism regarding insufficient baselines, we implemented several new methods.
To the best of our knowledge, differentiation through convex programs [2] is considered to be the ultimate solution for convex non-linear P\&O problems, as it computes the true gradient. Hence, other existing P\&O loss methods area built for the linear/combinatorial case.
We implemented SPO+ surrogate loss [1] (labeled ''SPO+'' in the figure), mean-squared error $\|\hat{w}-w\|^2_2$ (''MSE''), and perturbation-based approach [2] (''perturbed'').  and compared them to our $r-$smoothing method on the linear portfolio optimization problem ($\lambda=0$) and the OPF problem. The plots depict the test regret during training. In these experiments, our method significantly outperforms the baselines.

**Figure 2.** To provide additional evidence that the zero-gradient problem is indeed the reason why the $r-$smoothing and projection distance regularization outperform the standard approach in Figures 3,4 of the main paper, we measure two new metrics -- the norm of the loss function gradient and the number of active constraints. We plot the average of each of these quantities over the training dataset for each training epoch. Using the linear ($\lambda=0$) and quadratic ($\lambda=2$) portfolio optimization problems, we compare the $r-$smoothing and standard method (computing true Jacobian of the QP approximation by differentiating the KKT conditions). Panels (a-b) correspond to the linear problem, and (c-d) to the quadratic. It can be seen that the gradient norm indeed decreases rapidly for the standard method. Additionally, we see that the number of active constraints increases, in line with the theoretical results of Section 3.1.

**Figure 3.**
We also received questions regarding the generalizability of the QP approximation method, which we tried to address in Figure 3.
We could not find any benchmark for P\&O experiments that has a convex, nonlinear, and nonquadratic objective. Instead, we ran additional experiments with a slightly modified portfolio optimization problem. We substituted the linear term in the objective with the LogSumExp:
$$f(x, w, Q)= \log(\sum_i e^{w_ix_i}) - x^\top Q x.$$
This problem does not necessarily makes a lot of practical sense, but it allows us to test how well the QP approximation works when $f$ is not a quadratic function. In Figure 3 in the attached PDF, we compare the QP approximation without (labeled 'QP') and with (labeled '$r-$smoothing') our $r-$ smoothing technique to using the true function $f$ in the internal problem (labeled 'true $f$'). The results demonstrate that QP approximation performs even better than using the true problem. We believe that this is due to its simplicity --  the QP approximation only requires $n$ parameters to be predicted, while the true function $f$ has $n + n^2$ parameters.

[1] Adam N Elmachtoub and Paul Grigas. Smart predict, then optimize (2017)\
[2] Quentin Berthet et al. Learning with differentiable perturbed optimizers (2020)\
[3] Akshay Agrawal et al. Differentiable convex optimization layers (2019)\

---

### Decision · Program_Chairs · 2023-09-21

**Decision:**

Reject

**Comment:**

The paper studies the zero-gradient problem (the Jacobian has a large null space) in the predict-optimize framework. It proposes a method to solve this problem and empirically evaluates its approach on 2 real-world benchmarks.

After carefully reading the rebuttal/discussion, I do not think that the current version of the paper is ready for acceptance. Please incorporate the reviewers' feedback. In particular, addressing the following concerns will help strengthen the paper:

- Better situate the paper with respect to the previous work. For example, please refer and clearly compare to the past work (Elmachtoub and Grigas [2017] Wilder et al. [2019], Vlastelica et al.[2019] that acknowledge and address the zero-gradient issue.

- Better experimental evaluation and comparison to baselines that avoid the zero gradient problem by incorporating quadratic/exponential regularization terms, or those that consider the straight-through estimator (Sahoo et al, [2022]). Moreover, it is important to conceptually compare the proposed r-smoothing surrogate to previous work. For example, does the r-smoothing surrogate degenerate into adding a quadratic penalty in some special cases?